# A Retrospective Study of the Efficacy and Safety of Naldemedine for Treatment of Opioid-Induced Constipation in Patients with Hepatobiliary Pancreatic Cancer

**DOI:** 10.3390/medicina59030492

**Published:** 2023-03-02

**Authors:** Teruhiko Kamiya, Hisao Imai, Yukiyoshi Fujita, Eriko Hiruta, Takashi Masuno, Shigeki Yamazaki, Hajime Tanaka, Mitsuru Sandoh, Satoshi Takei, Kazuya Arai, Hiromi Nishiba, Junnosuke Mogi, Shiro Koizuka, Taeko Saito, Kyoko Obayashi, Kyoichi Kaira, Koichi Minato

**Affiliations:** 1Department of Pharmacy, Tatebayashi Kosei General Hospital, 262-1 Narushima, Tatebayashi 374-8533, Gunma, Japan; 2Division of Respiratory Medicine, Gunma Prefectural Cancer Center, 617-1 Takahayashi-nishi, Ota 373-0828, Gunma, Japan; 3Department of Respiratory Medicine, Comprehensive Cancer Center, International Medical Center, Saitama Medical University, 1397-1 Yamane, Hidaka 350-1298, Saitama, Japan; 4Division of Pharmacy, Gunma Prefectural Cancer Center, 617-1 Takahayashi-nishi, Ota 373-0828, Gunma, Japan; 5Division of Pharmacy, Fujioka General Hospital, 813-1 Nakagurisu, Fujioka 375-8503, Gunma, Japan; 6Division of Pharmacy, Kiryu Kosei General Hospital, 6-3 Orihime, Kiryu 376-0024, Gunma, Japan; 7Division of Pharmacy, Haramachi Red Cross Hospital, 698 Haramachi, Higashiagatsuma-machi, Agatsuma-gun 377-0882, Gunma, Japan; 8Division of Pharmacy, Ota Memorial Hospital, 455-1 Oshima, Ota 373-8585, Gunma, Japan; 9Division of Pharmacy, Tone Central Hospital, 910-1 Numasu, Numata 378-0012, Gunma, Japan; 10Division of Pharmacy, Gunma Saiseikai Maebashi Hospital, 564-1 Kamishinden, Maebashi 371-0821, Gunma, Japan; 11Division of Pharmacy, Japan Community Health Care Organization (JCHO) Gunma Chuo Hospital, 1-7-13 Kouun, Maebashi 371-0025, Gunma, Japan; 12Laboratory of Clinical Pharmacy, Faculty of Pharmacy, Takasaki University of Health and Welfare, 37-1 Nakaorui, Takasaki 370-0033, Gunma, Japan; 13Division of Pharmacy, Hidaka Hospital, 886 Nakao-machi, Takasaki 370-0001, Gunma, Japan; 14Division of Palliative Care, Gunma Prefectural Cancer Center, 617-1 Takahayashi-nishi, Ota 373-0828, Gunma, Japan

**Keywords:** efficacy, naldemedine, opioid-induced constipation, peripherally acting mu-opioid receptor antagonist, safety, hepatobiliary pancreatic cancer

## Abstract

*Background and Objectives*: Opioid analgesics, which are used for cancer-related pain management, cause opioid-induced constipation (OIC). Naldemedine, a peripheral opioid receptor antagonist, is an OIC-modifying agent, but no focused efficacy and safety analysis has been conducted for its use in hepatobiliary pancreatic cancers. We performed a multi-institutional study on the efficacy and safety of naldemedine in patients with hepatobiliary pancreatic cancer using opioids in clinical practice. *Materials and Methods*: We retrospectively evaluated patients with hepatobiliary pancreatic cancer (including liver, biliary tract, and pancreatic cancers) treated with opioids and naldemedine during hospitalization at ten institutions in Japan from June 2017 to August 2019. We assessed the frequency of bowel movements before and after the initiation of naldemedine therapy. Responders were defined as patients who defecated ≥3 times/week, with an increase from a baseline of ≥1 defecations/week over seven days after the initiation of naldemedine administration. *Results*: Thirty-four patients were observed for one week before and one week after starting naldemedine. The frequency of bowel movements increased by one over the baseline frequency or to at least thrice per week in 21 patients. The response rate was 61.7% (95% confidence interval: 45.4–78.0%). The median number of weekly bowel movements before and after naldemedine treatment was 2 (range: 0–9) and 6 (range: 1–17), respectively, in the overall population (n = 34); the increase in the number of bowel movements following naldemedine administration was statistically significant (Wilcoxon signed-rank test, *p* < 0.0001). Diarrhea was the predominant gastrointestinal symptom, and 10 (29.4%) patients experienced grade 1, grade 2, or grade 3 adverse events. The only other adverse event included fatigue in one patient; grade 2–4 adverse events were absent. *Conclusions*: Naldemedine is effective, and its use may be safe in clinical practice for patients with hepatobiliary pancreatic cancer receiving opioid analgesics.

## 1. Introduction

Hepatobiliary pancreatic cancer includes liver cancer, gallbladder cancer, biliary tract cancer, and pancreatic cancer [1,2,3]. Globally, the number of patients newly diagnosed with hepatobiliary pancreatic cancer in 2018 was approximately 1.85 million, accounting for 10% of all newly diagnosed cancer patients [4]. Hepatobiliary and pancreatic cancers have been associated with a poor prognosis owing to their high level of invasiveness, potential for distant metastasis, and resistance to conventional treatment modalities such as systemic chemotherapy [5,6,7]. Opioids are widely used as analgesic agents that play a central role in cancer pain management [8,9]. Opioids are effective for cancer pain treatment but often cause adverse events that negatively affect the patients’ quality of life; such adverse events frequently lead to the interruption of opioid therapy [10,11]. Opioid-induced constipation (OIC) is one of the most common side effects in patients receiving opioid treatment and is observed in more than 50% of such patients in the absence of preventive measures [12,13]. OIC is a type of functional constipation defined as altered bowel habits and defecation patterns after opioid administration [14]. The incidence varies depending on the type of opioid analgesic and the route of administration, but in general, opioid analgesics have been reported to have a high potential to cause OIC [15,16]. Unlike that for other adverse effects, such as nausea and vomiting, tolerance towards OIC is not likely to improve with the continued use of opioids [17]. Prolonged constipation has been reported to increase the incidence of delirium, nausea, vomiting, appetite loss, and abdominal pain [18]. These unpleasant adverse effects are major barriers to cancer pain management; they not only deteriorate the quality of life of patients but also lead to a decrease in the use of analgesics and rescue therapies [19]. Thus, addressing OIC in patients receiving opioids for the management of cancer-related pain is important. An earlier study reported that the cumulative incidence of OIC in Japanese patients with cancer was 56% in a week, as per the international Rome IV diagnostic criteria for OIC [20]. Furthermore, OIC and a decline in quality of life can occur rapidly in cancer patients after opioid administration [20]. Opioids have various effects on gastrointestinal function, leading to symptoms of dysphagia, nausea and/or vomiting, bloating, constipation, and abdominal pain. Thus, to effectively treat patients who are on chronic opioid therapy, physicians need to be able to identify these gastrointestinal adverse events and treat them appropriately [21]. In addition, opioids are known to cause the contraction of the sphincter of the Oddi muscle, which interferes with the expulsion of bile and pancreatic juice. Therefore, caution must be exercised when using opioids in patients with gallbladder disorders, cholelithiasis, or pancreatitis [21]. Patients with hepatobiliary and pancreatic cancers are expected to have reduced bile output; the contraction of the sphincter of Oddi induced by opioid administration may further reduce the output and decrease bile acid content, rendering the OIC more severe.

The primary mechanism by which opioids exert their analgesic effects is the activation of opioid receptors in the central nervous system. Stimulation of the μ-opioid receptors in the enteric tract inhibits intestinal peristalsis [22]; additionally, opioid-induced intestinal dysfunction develops relatively fast following the initiation of opioid administration [23]. Naldemedine is a peripheral-acting μ-opioid receptor antagonist (PAMORA) that reduces OIC by acting on opioid receptors in the gastrointestinal tract [24]. It recovers the gastrointestinal function by suppressing the effects of opioids on the intrinsic nervous system while preserving their analgesic effects. The phase III randomized trials COMPOSE-4 and COMPOSE-5 reported that naldemedine was efficacious and safe in cancer patients with OIC [25,26]. The most common toxicities reported were diarrhea (19.6%), fatigue (4.1%), appetite loss (3.1%), and vomiting (3.1%); 9.3% of patients were reported to discontinue treatment due to adverse events [25]. However, to date, those studies have been conducted on carefully selected patients with good performance status (PS) scores and adequate organ function, and no data have been reported on the efficacy and tolerability of naldemedine in patients with hepatobiliary pancreatic cancer. Although we previously evaluated the effectiveness and safety of naldemedine in combination with opioids in clinical practice [27,28], we did not adequately investigate its effectiveness in cancer patients with hepatobiliary pancreatic cancer. Furthermore, to date, there have been no reports focusing on the digestive system and hepatobiliary pancreatic cancer. Thus, the analysis of the effectiveness and toxicity of naldemedine for treating OIC in patients with hepatobiliary pancreatic cancers is important and warranted. We performed a multi-institutional retrospective analysis on the effectiveness and toxicity of naldemedine treatment in patients with hepatobiliary pancreatic cancer receiving opioids in clinical practice. Our results showed that naldemedine was efficacious in improving defecation frequency and was safe to use in clinical practice in hepatobiliary pancreatic cancer patients receiving opioid treatment.

## 2. Materials and Methods

### 2.1. Patients

This multi-institutional, retrospective study was conducted at ten institutions in Japan on patients with hepatobiliary pancreatic cancer treated with a combination of naldemedine and opioids between 7 June 2017, and 31 August 2019. Eligible patients who initiated naldemedine treatment during the target period were selected from pharmacy databases, and their clinical data were extracted from the electronic medical charts. Patients who met the following four criteria were included in the current analysis:

(i) cytologically or pathologically diagnosed with hepatobiliary pancreatic malignancies; (ii) naldemedine administration started during hospitalization; (iii) naldemedine administrated in combination with opioids; and (iv) hospitalized for at least seven days before and seven days after naldemedine treatment initiation. Data on the number of bowel movements were extracted from the medical records maintained by medical staff. First, we discerned 56 patients with hepatobiliary pancreatic cancer who were administered naldemedine for the first time in combination with an opioid during hospitalization. Of the eligible ones, 22 patients could not be followed for at least seven days before and seven days after the initiation of naldemedine and were therefore excluded. Overall, 34 patients were enrolled in this study (Appendix A). The data of the 34 patients included in this study have been previously published [27,28]. Patient charts were reviewed for data on patient characteristics, response to naldemedine, and adverse events. This study was approved by the Institutional Review Board of each participating institution. All procedures complied with the ethical standards of the institutional and/or national research committee and with the 1964 Declaration of Helsinki and its subsequent amendments or comparable ethical standards. Due to the retrospective nature of the study, the requirement for obtaining informed consent was waived, but the opportunity to refuse participation was guaranteed to eligible patients using an opt-out method.

### 2.2. Treatment

Cancer patients who had previously never received naldemedine were included. The treatment comprised an oral intake of 0.2 mg naldemedine once daily with an opioid. The administration of naldemedine was continued until the development of intolerable adverse events, withdrawal of consent of treatment, or until the physician determined that discontinuation was necessary. The decision to initiate and terminate naldemedine administration was made at the discretion of the attending physician.

### 2.3. Assessment of Therapeutic Efficacy

Information on the weekly frequency of bowel movements (times/week) for the week prior to and the week following the initiation of naldemedine administration was extracted from the medical records. A responder was defined as a patient with ≥3 bowel movements/week in the first seven days following initiation of naldemedine administration and who exhibited an increase of at least one bowel movement from the baseline value. Baseline defecation frequency was defined as the number of bowel movements during the seven days prior to the initiation of naldemedine administration. Adverse events were assessed according to the Common Terminology Criteria for Adverse Events version 5.0.

### 2.4. Statistical Analysis

Fisher’s exact test was used to analyze categorical variables. After checking for normality and equal variances, we adopted the Wilcoxon signed-rank test to evaluate the correspondence between the two groups. Statistical analysis was performed with multivariate ordered logistic regression analysis to determine factors that predict treatment effectiveness. The results obtained from the multivariate ordered logistic regression analysis were presented as odds ratios and 95% confidence intervals. Statistical significance was defined based on a two-sided *p*-value ≤ 0.05. All statistical analyses were performed using the JMP for Windows software, version 12.0 (SAS Institute, Cary, NC, USA).

## 3. Results

### 3.1. Patient Backgrounds

Of the 34 patients enrolled in this study, 28 had died of the underlying disease by the data-cutoff time. The background characteristics of the patients are listed in Table 1. The median age of the patients in the analysis was 72 years (range, 43–87 years), and 13 (38.2%) patients were aged ≥75 years. Regarding sex, 16 patients were men and 18 were women. Based on the Eastern Cooperative Oncology Group Performance Status (ECOG-PS) criteria, the PS scores were as follows: 11 (32.4%) patients, PS 0 or 1; 7 (20.6%) patients, PS 2; and 16 (47.0%) patients, PS 3 or 4 (considered to be a poor PS). The distribution of cancer types was as follows: 23 (67.7%) patients had pancreatic cancer, 8 (23.5%) had liver cancer, and 3 (8.8%) had cholangiocarcinoma of the gallbladder and bile ducts.

Table 2 presents the data on the usage of opioids, laxatives, and antiemetic drugs in the current study population. The median regular opioid dose in oral morphine equivalents was 30 mg/day (range: 7.5–600 mg). With regard to opioid use, the most commonly administered opioid was oxycodone, which was administered to 20 patients (58.8%), followed by fentanyl, which was administered to 11 patients (32.4%). Regarding the use of laxatives, 24 (70.6%) patients received concomitant laxatives, of which 21 (87.5%) received magnesium oxide; sennoside and lubiprostone were administered to 4 (11.8%) patients each. Furthermore, the median value for days from the first opioid treatment to the initiation of naldemedine administration was five days (range, 1–287 days); 15 (44.1%) patients initiated naldemedine treatment within 14 days of the initiation of opioid administration. With regard to antiemetic medications, 15 patients (44.1%) received concomitant antiemetic medications (regular or occasional use), with prochlorperazine being the most commonly used antiemetic agent (8 patients; 53.3%).

### 3.2. Treatment Efficacy and Safety

The frequency of bowel movements was examined for at least seven days before and seven days after the initiation of naldemedine treatment in all 34 included patients. The number of responders and non-responders is shown in Figure 1; 21 (61.7%, 95% CI: 45.4–78.0%) patients were responders, and 13 were non-responders. Table 3 shows the patient background information based on the response. The background characteristics of the responders and non-responders were not significantly different as per Fisher’s exact test.

In addition, changes in the weekly frequency of bowel movements in the week prior to and the week after the start of naldemedine treatment were examined for the following patient cohorts: all 34 patients and those who had less than three bowel movements in the seven days prior to naldemedine administration (Figure 2). In all the eligible patients (n = 34), the median number of bowel movements that occurred in the seven days before and seven days following naldemedine treatment was 2 (range: 0–9) and 6 (range: 1–17), respectively; hence, the number of bowel movements increased significantly following naldemedine initiation (*p* <0.0001; Figure 2a). Furthermore, we compared the weekly number of bowel movements during the week prior to and the week following naldemedine initiation in patients who had less than three bowel movements in the week prior to naldemedine treatment (n = 19). The median number of bowel movements that occurred during the seven days before and seven days following naldemedine initiation was 1 (range, 0–2) and 3 (range, 0–7), respectively; hence, the number of bowel movements increased significantly following naldemedine initiation (*p* = 0.0001; Figure 2b).

As listed in Table 4, diarrhea was the most common adverse event likely associated with naldemedine, which was reported (regardless of grade) in 10 patients (29.4%); of these, nine (90.0%) patients reported grade 1 or 2 diarrhea. None of the patients experienced adverse events of grade 4 or higher.

### 3.3. Clinical Factors Influencing Treatment Response

Finally, multivariate logistic regression analysis was performed to assess the relationship between naldemedine efficacy and various clinical factors, as shown in Table 5. There were no statistically significant differences in the efficacy of naldemedine with respect to age, PS, or daily opioid dose in oral morphine equivalents.

## 4. Discussion

In the current analysis, we assessed changes in the number of bowel movements and toxicities after naldemedine initiation in patients with hepatobiliary pancreatic cancer who were treated with opioids and hospitalized for at least seven days before and seven days following initiation of naldemedine administration. Additionally, we examined the effects of various associated factors that may affect treatment response.

In the current study, the response rate was 61.7%, which is comparable to those reported by the COMPOSE-4 study (71%) and another clinical trial on naloxegol (73%), a PAMORA drug similar to naldemedine [25,29]. However, patient characteristics differed between the previous studies and the present study, and thus the results of the studies cannot be compared. Furthermore, in the present study, a significant increase in the number of bowel movements was found following naldemedine administration in the entire study population. In addition, a statistically significant increase in the number of bowel movements occurred in the patients who had fewer than three bowel movements in the week prior to naldemedine treatment. Thus, the present results suggest that naldemedine is effective for hepatobiliary pancreatic cancer patients with OIC. As per our multivariate logistic regression analysis, none of the clinical factors we examined (age, PS, or daily opioid dose in oral morphine equivalents) were significant with reference to the effectiveness of naldemedine. These results are similar to those of previous studies that have reported that the efficacy of naldemedine in patients with OIC is independent of the baseline characteristics of patients [30,31].

In our study, 47.0% of patients exhibited PS of ≥3, whereas the COMPOSE-4 and COMPOSE-5 prospective studies of naldemedine in patients with OIC included cancer patients with PS 0–2 and excluded patients with PS ≥ 3 [25]. Thus, the effectiveness and tolerability of naldemedine in clinical practice have not been assessed in prospective studies. Additionally, there are no prospective clinical trials focused specifically on patients with hepatobiliary pancreatic cancer. Furthermore, although naldemedine is prescribed to many outpatients in real-world clinical settings, the accurate assessment of the number of bowel movements in such patients in practice is difficult; thus, the data in the current study were limited to inpatients. Inpatient data are thought to be more reliable because they are recorded and assessed by healthcare professionals. Hospitalization for at least seven days before and seven days following the initiation of naldemedine treatment was required to gather sufficient data for the comparative analysis. However, the results should be interpreted with caution because the patients may have been hospitalized due to complications, such as poor PS, or because they required concomitant therapy.

In terms of tolerability, diarrhea and abdominal pain were the most frequently occurring toxicities, with incidence rates ranging from 19.6% to 39.7% and 1.7%, respectively. This is in agreement with the results from other prospective clinical studies in cancer patients with OIC [25,32]. The occurrence of diarrhea and abdominal pain in the current analysis population was 29.4% and 0%, respectively, almost equal to that in the randomized phase III study [25,32]. Despite the fact that the study population comprised patients with PS 3–4 and those aged 75 years and over, only one serious adverse event (2.9%, grade 3 diarrhea) was observed, suggesting that naldemedine treatment was well tolerated in hepatobiliary pancreatic cancer patients in clinical practice.

A study on a variety of malignancies, including gastrointestinal cancers and other abdominal organs, has shown results comparable to those of the COMPOSE-4 and -5 phase III studies with regard to the effectiveness and toxicities of naldemedine [25]. However, patient selection bias was inherent in the cited study as only naldemedine-eligible patients without gastrointestinal obstruction who were able to receive naldemedine orally were enrolled. This study included patients with hepatobiliary pancreatic cancer in clinical practice; however, unlike the patients enrolled in the clinical trial, not all patients had a good general condition or were complication-free, but naldemedine administration was possible because they were able to take it orally. In terms of age, almost one-third of cancers are diagnosed in patients aged 70 years or over; this poses additional challenges with respect to the optimal therapeutic approach and prognosis in this patient population [33]. In general, older cancer patients exhibit a higher number of comorbidities and reduced organ function compared to younger cancer patients; therefore, treatment-related adverse events in geriatric patients is a noteworthy concern. While geriatric patients are usually excluded from randomized phase III studies, COMPOSE-4 and -5 included patients aged 20 years and older, and no upper age limit was stipulated [25]. In addition, a subgroup analysis of a phase III study in patients with chronic non-cancer-related pain demonstrated that naldemedine treatment was generally effective and tolerable, even in patients aged 65 years and older [34]. Similar to this analysis, the current study found no statistically significant difference in the effectiveness of naldemedine in patient groups older and younger than 75 years, showing that naldemedine may be effective in treating elderly patients with hepatobiliary pancreatic cancer. Opioids affect the flow of bile in the bile ducts, and the current evidence suggests that increasing the frequency of contraction of the sphincter of Oddi reduces the filling of the peristaltic segments of the bile ducts and decreases the emptying of the bile ducts [35]. The mechanism appears to involve the suppression of inhibitory motoneurons controlling the sphincter muscle. Opioids also decrease the contractility of the gallbladder. In a previous animal study, opioid agonists inhibited excitatory neurotransmission at the presynaptic and neuromuscular terminals [36]; moreover, a human study reported a decrease in biliary ejection fraction in 76% of patients who received opioids followed by cholecystokinin [37], potentially predisposing the patients to gallbladder stasis. The latter study also reported an increased flow of hepatic bile toward the gallbladder rather than the small intestine (such as that in a fasting state), which may be due to the effect of opioid administration on the sphincter of Oddi [37]. In patients with hepatobiliary pancreatic cancer, the ability to expel bile is expected to be reduced, and the contraction of the sphincter of Oddi induced by opioid administration may further reduce bile acid expulsion [21]; this may increase the severity of OIC. It is therefore important to manage OIC adequately in these patients. Although there have been no reports on the effectiveness and tolerability of naldemedine in gastrointestinal cancers, we previously reported on the effectiveness and tolerability of naldemedine in thoracic cancers [38]. The results showed that 65.0% of patients were responders, and the safety profile was comparable to that described in the present study in hepatobiliary pancreatic cancer; diarrhea occurred in 27.5% of the patients. The above-mentioned results imply that no special considerations need to be taken into account for patients with hepatobiliary pancreatic cancer with reference to the efficacy and adverse events of OIC treatment; such treatment can be applied in the same manner as in other malignancies.

Several limitations exist in the current study. First, the population analyzed in this study is small. However, the study has clinical significance because the inclusion criteria specified hospitalized hepatobiliary pancreatic cancer patients whose number of bowel movements was closely followed and evaluated by healthcare professionals for at least seven days before and seven days following naldemedine administration. Second, owing to the retrospective nature of the study, objective evaluations, such as the Bristol stool form scale [39], bowel function index [40], and defecation diary, were not applicable. This study limitation may affect the validity of our results. Third, the decision to start, skip, or discontinue naldemedine administration and the use of various laxatives was at the discretion of each attending physician, which may have introduced discrepancies due to subjectivity. Fourth, owing to the nature of this retrospective analysis, standardizing the effects of other concomitant medications was not possible.

## 5. Conclusions

In summary, the results of the current study indicate that naldemedine treatment for hepatobiliary pancreatic cancer patients is effective and well tolerated in clinical practice; this drug can be administered to elderly patients and patients with poor PS. Overall, naldemedine is effective and well tolerated for OIC in most hepatobiliary pancreatic cancer patients.

## Figures and Tables

**Figure 1 medicina-59-00492-f001:**
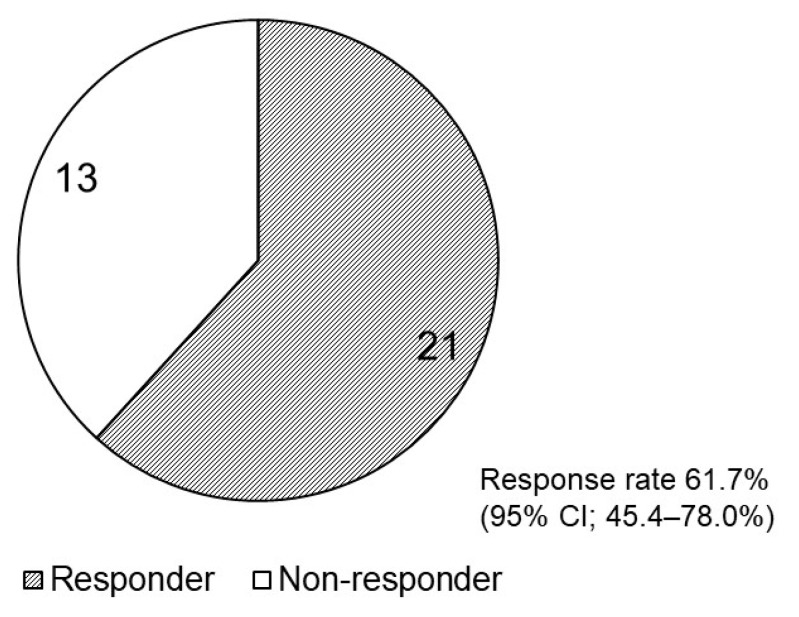
Pie chart presenting the number of responders and non-responders after naldemedine initiation. CI, confidence interval (CI).

**Figure 2 medicina-59-00492-f002:**
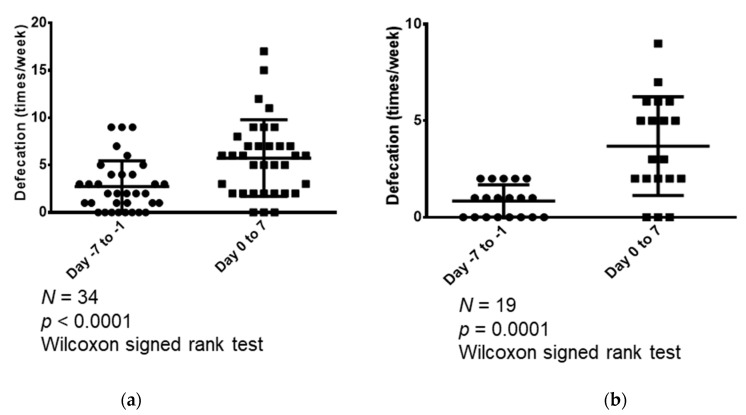
Comparison of the weekly number of bowel movements in the week (seven days) prior to and the week following naldemedine initiation. (**a**) Comparison of the number of bowel movements that occurred during the seven days before and seven days following naldemedine initiation in all patients (n = 34); (**b**) Comparison of the number of bowel movements that occurred during the seven days before and seven days following naldemedine initiation in patients who had less than three bowel movements in the seven days prior to naldemedine administration (n = 19).

**Table 1 medicina-59-00492-t001:** Baseline patient characteristics.

Characteristic	N = 34
Sex	
Men/Women	16/18
Median age upon the initiation of naldemedine treatment (years) (range)	72 (43–87)
Performance status	
0/1/2/3/4	4/7/7/15/1
Primary tumor	
Liver cancer	8
Cholangiocarcinoma of the gallbladder and bile ducts	3
Pancreatic cancer	23
Therapy before and during naldemedine initiation *	
Systemic anticancer drugs	7
Abdominal irradiation	0
Abdominal surgery	1
Best supportive care alone	26
History of abdominal surgery before the initiation of naldemedine	
Yes/No	10/24
History of irradiation to the abdomen and pelvic region before the initiation of naldemedine	
Yes/No	2/32
Central nervous system metastasis including brain metastasis and carcinomatous meningitis	
Yes/No	0/34
Carcinomatous peritonitis	
Yes/No	6/28
Gastrointestinal obstruction	
Yes/No	0/34
Presence of diabetes mellitus	
Yes/No	6/28
Discontinuation within one week	
Yes/No	4/30
Use of laxatives before the initiation of naldemedine	
Yes/No	27/7
Concomitant use of laxatives during naldemedine treatment	
Yes/No	24/10
Regular use of antiemetic medications after the initiation of naldemedine	
Yes/No or unknown	13/21
Occasional use of antiemetic agents after the initiation of naldemedine	
Yes/No or unknown	5/29
Survival status at data-cutoff time	
Dead/Alive	28/6
Period to death from the beginning of naldemedine treatment	
Median period (days) (range)	41.5 (9–407)

* Within 3 weeks before the initiation of naldemedine.

**Table 2 medicina-59-00492-t002:** Opioids, laxatives, and antiemetic drugs used in this study.

Opioids, Laxatives, and Antiemetics		%
Daily opioid dose in oral morphine equivalents (mg)		
Median (range)	30 (7.5–600)	
<30	14	41.2
30–99	12	35.3
≥100	8	23.5
Regular use of opioids (no. of patients)		
Oxycodone	20	58.8
Fentanyl	11	32.4
Morphine	2	5.9
Hydromorphone	1	2.9
Days from the first opioid initiation to naldemedine treatment (days)		
Median (range)	5 (1–287)	
<4	9	26.5
4–7	3	8.8
8–14	3	8.8
15–28	7	20.6
29–99	9	26.5
≥100	3	8.8
Concomitant laxatives * (no. of patients)		
Magnesium oxide	21	61.8
Sennoside	4	11.8
Lubiprostone	4	11.8
Bisacodyl	2	5.9
Sodium picosulfate hydrate	2	5.9
Sodium bicarbonate, sodium dihydrogen phosphate anhydrous suppository	1	2.9
Others	2	5.9
Concomitant antiemetic drugs (regular and occasional use) * (no. of patients)		
Prochlorperazine	8	23.5
Metoclopramide	5	14.7
Domperidone	3	8.8
Olanzapine	2	5.9
Others	0	0

* Includes duplicate use.

**Table 3 medicina-59-00492-t003:** Patient backgrounds based on response.

Characteristic	Responder (n = 21)	Non-Responder (n = 13)	*p*-Value
Sex			
Men/Women	8/13	8/5	0.29
Age			
<75/≥75	13/8	8/5	>0.99
PS			
0–2/≥3	12/9	6/7	0.72
Regular opioid dose (morphine equivalent; mg/day)			
<30/≥30	7/14	7/6	0.29
History of chemotherapy within 21 days prior to naldemedine initiation			
Yes/no	6/15	1/12	0.21
History of abdominal surgery before naldemedine initiation			
Yes/No	9/12	1/12	0.05
History of radiotherapy to the abdomen and pelvic region before naldemedine initiation			
Yes/No	2/19	0/13	0.51
Presence of diabetes mellitus			
Yes/No	3/18	3/10	0.65
Use of laxatives before the initiation of naldemedine			
Yes/No	16/5	11/2	0.68
Concomitant use of laxatives during naldemedine treatment			
Yes/No	17/4	7/6	0.12

PS, performance status.

**Table 4 medicina-59-00492-t004:** Toxicities during naldemedine treatment.

Adverse Events *	Gr. 1	Gr. 2	Gr. 3	Gr. 4	Total
Diarrhea	7	2	1	0	10
Abdominal pain	0	0	0	-	0
Nausea	0	0	0	-	0
Vomiting	0	0	0	0	0
Anorexia	0	0	0	0	0
Fatigue	1	0	0	-	1

* Adverse events were graded based on the Common Terminology Criteria for Adverse Events version 5.0; Gr, Grade.

**Table 5 medicina-59-00492-t005:** Multivariate logistic regression analysis of the clinical factors indicative of response in patients treated with naldemedine.

Variables	Odds Ratio	95% CI	*p*-Value
Age (years)			
<75/≥75	1.31	0.25–7.36	0.74
PS			
0–2/≥3	0.38	0.05–2.03	0.26
Daily opioid dose in oral morphine equivalents			
<30/≥30	3.55	0.66–23.3	0.13

CI, confidence interval; PS, performance status.

## Data Availability

The data presented in this study are available on request from the corresponding author. The data are not publicly available.

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
