# Peer review of "A Retrospective Study of the Efficacy and Safety of Naldemedine for Treatment of Opioid-Induced Constipation in Patients with Hepatobiliary Pancreatic Cancer"

_medicina, 2023, doi:10.3390/medicina59030492_

Round 1

Reviewer 1 Report

Very importatnt topic since OIC is affecting QoL very much. Standard laqatives are quite often not effective enough and therefore introduction of new effective drug is more than needed. Strengh of this study is "real practice" use - in a group of cancer patients with not good performace status. 

Author Response

Response: We greatly appreciate you taking time out of your busy schedule to review our manuscript. Your valuable comments have encouraged us to reaffirm the value of our research.

Reviewer 2 Report

I attached my report.

Author Response

Responses to Reviewer 2 Comments:

Review report

This represents an interesting paper on the role of naldemedine in a particular class of patients receiving palliative care due to hepato-biliary and pancreatic cancer. The data reported from a multicentric analysis are supportive on the role of this drug in reducing OIC. However some more explanations should be given to confer an even more value to this paper.

Response: We greatly appreciate you taking time out of your busy schedule to review our manuscript. We have implemented the required modifications in the manuscript as per your suggestions.

INTRODUCTION:

The order of concepts explained in the introduction section should be revised since the definition of OIC was given only at row 84-85 and maybe could be useful to have it at the beginning.

Response: Thank you for your insightful comment. As per your suggestion, we moved the OIC definition to an earlier position (page 2, lines 76–77).

RESULTS section

Patients analyzed were on several different laxatives and on some antiemetic drugs known to have a pro-peristaltic role. How to justify this possible risk of bias?

Can diarrhea be explained by not just the intake of naldemedine but from a combination of this with other drugs?

The multivariate analysis should consider all these factors.

Response: Thank you for pointing this out. It is not possible to completely rule out bias regarding the effects of several laxatives and antiemetics. This is a limitation of retrospective studies. Also, for a multivariate analysis, we believe that including three factors at most is reasonable for a cohort of 34 patients. We will take your suggestion into consideration in future studies with a larger scale that are not limited to hepatobiliary pancreatic cancers.

Moreover, speaking about HPB patients, although each of them on palliative care, some previous surgery background may modify the intestinal symptoms. Did any of them receive any sort of resection or any palliative bypass?

Since the unicity of the study in focusing on such a specific class of patients, the value of the work produced would be even increased by reporting some clinical details of the analyzed group.

Response: Thank you for your valuable comment. In our analysis, we were not able to examine the details of surgical procedures and only took into consideration the history of abdominal surgery before naldemedine initiation. In future studies, we aim to take your suggestion into consideration by extracting the data pertaining to surgical techniques when investigating hepatobiliary pancreatic cancers.

An attempt of sub-analysis should be made. Which are the differences between patients responding and not?

Response: Thank you for your valuable comment. Table 3 compares responders and non-responders, and no differences were observed between the two groups in terms of the analyzed factors. I think a limitation of the subanalysis is the relatively small number of patients (n=34), which limits the comparison to only two groups, responders and non-responders. We will gladly take your insightful suggestion into consideration when conducting studies with larger cohorts in the future.

DISCUSSION

Row 278-279 Rephrase. Although statistically supported, the results of a retrospective small group of patients may only suggest and not demonstrate the efficacy of this drug. Prospective analyses are needed.

Response: Thank you for your valuable comment. As per your suggestion, we have rephrased the sentence as follows:

“Thus, the present results suggest that naldemedine is effective for hepatobiliary pancreatic cancer patients with OIC.”

Row 312-313 Rephrase. Which are the advantages in terms of patient selection bias of this study compared to others reported?

Response: Thank you for your valuable comment. As per your suggestion, we have rephrased the sentence as follows:

“However, patient selection bias was inherent in the cited study as only naldemedi-ne-eligible patients without gastrointestinal obstruction who were able to receive naldemedine orally were enrolled. This study included patients with hepatobiliary pancreatic cancer in clinical practice; however, unlike the patients enrolled in the clinical trial, not all patients had a good general condition or were complication-free, but naldemedine administration was possible because they were able to take it orally.”

In conclusion, the analyses along with the data reported has been well conducted even if the population (apart from the retrospective analysis) may carry several background differences. Clearly, considering the small number of patients, a matching cannot be applied, and other most sophisticated statistical analysis should be attempted (as an inverse probability weighting). Anyway, at least differences should be explicated, clearly reported and considered as a limit other than the greatest potential risk of bias of this study.

Response: Thank you for your insightful comment. As you pointed out, a limitation of this study is the small sample size. To address your concern, we included the following statement in the study limitations paragraph.

“First, the population analyzed in this study is small. However, the study has clinical significance because the inclusion criteria specified hospitalized hepatobiliary pancreatic cancer patients whose number of bowel movements was closely followed and evaluated by healthcare professionals for at least seven days before and seven days after naldemedine administration.”

Reviewer 3 Report

Dear Authors

Thank you for this manuscript. OIC is one of the most common and important adverse events of opioids, influencing the effectiveness of cancer pain treatment and overall patients quality of life. Hepatobiiliary and panreatic cancers are not the most common ones, but occurence of multiple gastrointestinal symptoms and sometimes swift increase in opioid doses due to fast aggreviation of pain intensity (pancreatic cancer specially) make the problems with constipation of highest importance in that group of patients.

The study disagn and statistics used for the study are appropriate to me.

The results are correcty, clearly and understandably described, followed by proper discussion.

All the important study limitations are outlined and described.

Based on so far available datas naldemedine seems to be one of the important drugs to deal with OIC or even more widely with OIBD with very good responce rate and sfety for patients.

Author Response

(The authors gave the same response as above.)

Round 2

Reviewer 2 Report

Thank you for the reply to my comments and for the improvements made. Although some important limitations still exist, this represent a valuable work to be published